# Barriers to Optimal Child Sleep among Families with Low Income: A Mixed-Methods Study to Inform Intervention Development

**DOI:** 10.3390/ijerph20010862

**Published:** 2023-01-03

**Authors:** Elizabeth L. Adams, Amanda Edgar, Peyton Mosher, Bridget Armstrong, Sarah Burkart, R. Glenn Weaver, Michael W. Beets, E. Rebekah Siceloff, Ronald J. Prinz

**Affiliations:** 1Department of Exercise Science, University of South Carolina, 921 Assembly Street, Columbia, SC 29208, USA; 2Research Center for Child Well-Being, University of South Carolina, 1400 Pickens Street, Suite 400, Columbia, SC 29201, USA; 3Department of Psychology, Barnwell College, University of South Carolina, 1512 Pendleton Street, Columbia, SC 29208, USA

**Keywords:** child sleep, preschool children, intervention, parenting, low-income, prevention

## Abstract

This study gathered formative data on barriers to optimal child sleep to inform the development of a sleep intervention for parents of preschool-aged children in low-income households. Parents (*n* = 15, age: 34 ± 8 years, household income: $30,000 ± 17,845/year) reporting difficulties with their child’s sleep participated in this study. Mixed methods included an online survey and semi-structured phone interview. Items assessed barriers/facilitators to optimal child sleep and intervention preferences. Interview transcripts were coded using inductive analyses and constant-comparison methods to generate themes. Derived themes were then mapped onto the Theoretical Domains Framework to contextualize barriers and inform future intervention strategies. Themes that emerged included: stimulating bedtime activities, child behavior challenges, variability in children’s structure, parent work responsibilities, sleep-hindering environment, and parent’s emotional capacity. Parent’s intervention preferences included virtual delivery (preferred by 60% of parents) to reduce barriers and provide flexibility. Mixed preferences were observed for the group (47%) vs. individual (53%) intervention sessions. Parents felt motivated to try new intervention strategies given current frustrations, the potential for tangible results, and knowing others were in a similar situation. Future work will map perceived barriers to behavior change strategies using the Behavior Change Wheel framework to develop a parenting sleep intervention.

## 1. Introduction

Adequate sleep is essential to multiple domains of child development. Suboptimal sleep is linked with a range of detrimental child health outcomes, such as increased cardiometabolic risk and lower diet quality, as well as poor cognition and lowered psychosocial well-being [1,2,3]. Currently, ~33% of preschool-aged children do not receive the recommended amount of sleep per day, while ~50% of children in poverty do not obtain sufficient sleep [4]. Thus, interventions aimed at reducing these striking sleep disparities are crucial for sleep health equity and disease prevention. 

Early childhood is an optimal time for the primary prevention of suboptimal sleep. Social, cognitive, and physical competencies that are important for adaptive functioning emerge rapidly during early childhood and are supported by healthy sleep patterns [5]. At the same time, bedtime resistance, night wakings, co-sleeping, and variability in daytime naps are common in young children and impact overall sleep health [6]. These factors warrant important developmental considerations when designing interventions to promote optimal sleep during the preschool years [7]. Despite the importance of healthy sleep habits during this period of development, preventive interventions to improve sleep in early childhood are lacking. Existing sleep interventions have either focused on infancy, targeted clinical sleep disorders, or included sleep as one aspect of a multi-component intervention targeting several health behaviors [8]. Thus, it remains unknown if targeting sleep alone during early childhood results in improvements in related health domains (e.g., diet) that collectively prevent later-life cardiometabolic disease risk.

Few behavioral sleep interventions have also included families with low income, for whom preventive interventions may be especially beneficial [8,9]. Thus, to diversify the populations served, the inclusion of families with fewer resources should be prioritized using a purposeful approach that recognizes their unique challenges and needs. For example, thoughtful intervention design should consider multiple levels of influence on children’s sleep, including the bedroom, home, neighborhood, and daycare environments [7]. Although various individuals within these settings may influence sleep patterns, parents have a central role in fostering the development of young children’s healthy sleep patterns and routines [10]. As such, research must engage parents as key stakeholders by incorporating their viewpoints—and prioritizing their preferences—when developing interventions to support families.

As a first step, formative evidence is essential to understanding the various factors that influence sleep patterns across multiple domains in early childhood. This foundational evidence is critical to informing the design and delivery of effective interventions and maximizing the potential for their success [11]. Developmental evaluation is the first stage of the formative process and gathers community input on contextual barriers to the target behavior, as well as perceptions on the feasibility of implementation strategies for intervention delivery [12]. Currently, formative data on children’s sleep are limited to infants [13] and older populations [14], or distinct conditions (e.g., cerebral palsy) [15], specific locations (e.g., hospital settings) [16], and singular ethnic groups (e.g., Latinx) [17], resulting in evidence gaps that warrant further evaluation.

The aim of this study was to conduct a mixed-methods formative evaluation among parents with low income to identify perceived barriers to optimal child sleep in preschool-aged children. Parents’ motivations for participating in future interventions and their preferences for intervention delivery were also explored. These data will be used to inform intervention development to improve child sleep in under-resourced families, with assessments on outcomes across multiple health behaviors (e.g., diet) linked to cardiometabolic disease risk. Given the exploratory nature of this study design, no a priori hypotheses were specified.

## 2. Materials and Methods

### 2.1. Study Design 

This mixed-methods study used a concurrent triangulation design [18]. Priority was placed on qualitative data, and complementary information was provided in a quantitative context. Upon enrollment, parents completed an online survey, followed by a semi-structured phone interview. All study procedures were approved by the university’s Institutional Review Board.

### 2.2. Participant Recruitment

Parents (*n* = 15) were recruited between March and June 2022. The sample size was selected based on principles of qualitative research as an estimate for reaching saturation and sufficiency in the interview data [19]. Flyers were posted in pediatric clinics, in childcare newsletters, and on childcare social media sites. Interested parents were directed to a study webpage where they completed an eligibility screener. Parents had to (1) be ≥18 years of age; (2) have a child 2–4 years of age; (3) be a primary caregiver (i.e., child lived in their home ≥50% of the time); (4) earn a household income <200% of the Federal Poverty Line [20]; (5) report their child had trouble sleeping [21]; and (6) express interest in improving their child’s sleep. Parents were excluded if their child had a diagnosed sleep disorder or medical condition that significantly impacted their child’s sleep (e.g., epilepsy, sleep apnea). Primary caregivers who were not biological parents (e.g., grandparents; aunt) were considered eligible; however, only mothers and fathers enrolled in this study. For multi-parent households, only the interested parent who completed the study screener was interviewed. This sample was selected to represent parents with lower income, who would likely enroll in a behavioral sleep intervention for subclinical child sleep challenges. 

### 2.3. Quantitative Survey 

Upon enrollment, parents completed an online survey, lasting ~20 min. Survey items included demographics, the home environment, child sleep quality, current sleep guidance, and intervention preferences. The validated Child Sleep Wake Scale (CSWS) [22] was used to contextualize child sleep quality. The CSWS consists of 25 items, with response options rated on a 6-point scale, ranging from “Never” to “Always”. Five subscales (going to bed, falling asleep, maintaining sleep, reinitiating sleep, and returning to wakefulness) and a total sleep quality score were derived from select items [22]. Possible scores ranged from 1 to 6, with higher scores indicating better sleep quality in each domain. Survey items related to the home environment, parent’s receipt of sleep guidance, and intervention preferences were developed by the research team. These items assessed factors such as household level noise, presence of screens (e.g., phone, television) in their child’s bedroom, current sources of sleep guidance, sleep challenges parents struggle with most, parents’ interest in a child sleep intervention, preferred delivery mode, and ideal intervention length. 

### 2.4. Qualitative Interviews

A semi-structured interview guide was developed to elicit parents’ perceived barriers/facilitators to child sleep, and motivations/preferences for future intervention delivery. Interview questions addressed various aspects of sleep (i.e., nighttime sleep, night wakings, morning routines, naps), environmental contexts (i.e., bedroom, home, and neighborhood environments), salient challenges, motivations to try new intervention strategies, and intervention preferences. The interview guide was refined through an iterative process of peer debriefings and preliminary interviews until a final guide was reached (See Appendix A) [23]. Two researchers (ERS, ELA), trained in qualitative methodology, conducted practice phone interviews with parents not enrolled in this study (*n* = 4) [23]. Parent feedback was elicited to strengthen the clarity of the interview questions and enhance alignment with the research aims. The same two researchers then conducted primary interviews with enrolled parents that lasted ~40 min each. Interviews were audio-recorded and transcribed using an online transcription planform (Otter.ai; accessed on 2 January 2022).

### 2.5. Conceptual Approach 

The Theoretical Domains Framework (TDF) [24] was used to contextualize perceived barriers from the qualitative interviews. TDF was developed through a synthesis of 33 theories on behavior change, which distilled 84 theoretical constructs into 14 clustered domains [25]. These 14 domains provide a validated taxonomy of implementation determinants across cognitive, affective, social, and environmental influences for a target behavior. For example, the TDF domain of *behavioral regulation* pertains to barriers around self-monitoring, breaking a habit, and action planning, while the domain of *environmental context and resources* pertains to barriers in the environment and available resources. TDF was chosen as a guiding framework, given its extensive evidence base [26], and its ability to guide future intervention development by mapping behavior change strategies to associated barriers using the Behavior Change Wheel Framework [27]. 

### 2.6. Data Analysis

#### 2.6.1. Quantitative Analyses

Descriptive statistics were used to analyze quantitative survey items. Continuous variables were tested for normality using the Shapiro–Wilk test. Most variables were normally distributed, with the exception of child age, and CSWS variables for the amount of time children resisted bedtime, the amount of time to fall back to sleep after waking in the night, and time to become alert in the morning. As such, all continuous variables were reported as means ± standard deviations and median (interquartile range [IQR]). Categorical variables were reported as percentages. Analyses were conducted in SAS Studio version 3.8 (SAS Institute Inc., Cary, NC, USA).

#### 2.6.2. Qualitative Analyses

Interview transcripts were imported into NVIVO version released in March 2020 (QSR International, Burlington, MA, USA). Authors (ELA, AE, PM) trained in qualitative methodology generated themes using inductive analysis and an immersion crystallization approach [28,29,30]. Aligning with the phases of thematic analysis [31], two coders (AE and PM) read all transcripts and independently generated initial codes using recurring words, phrases, and themes. Coders then met with a third reviewer (ELA) to integrate and add codes into a comprehensive coding guide and settle any discrepancies. Subsequent transcripts were reviewed to determine if additional codes were needed. This iterative process was repeated until all transcripts were read, and coders agreed saturation had been reached using a “code meaning” approach [32,33]. Saturation reached transcript #11. Themes were then generated via collaborative discussions among coders using a constant-comparison method [28] and independently verified by all coders to ensure data consistency and integrity. Finalized themes on barriers to optimal child sleep were then mapped onto the TDF, as described above. 

#### 2.6.3. Trustworthiness of Findings

Prior to data collection, all coders engaged in a reflective process to examine personal biases and assumptions associated with this work [34]. Personal value systems were explored and areas for potential role conflict were identified. This process resulted in written positionality statements to acknowledge personal subjectivities [34]. During the phone interviews, rapport was established at the onset, and parents were encouraged to be honest and frank. Parents were told they were not required to share information they preferred to keep private, and iterative questioning was used to clarify contradictions for greater transparency [35]. To establish clarity of the research findings, peer scrutiny was conducted with a researcher not involved in transcript coding or theme generation (ERS). Their feedback was incorporated to modify and strengthen theme development. Frequent debriefing sessions were held during the analysis phase to resolve disagreements and achieve a consensus on the interpretation of participant responses [35]. Lastly, quantitative data were triangulated with the qualitative findings to result in a holistic understanding of the research questions [18].

## 3. Results

### 3.1. Demographics, Home Environment, and Child Sleep Characteristics 

Parent and child demographics are listed in Table 1. About half (53.3%) of children had access to screens in their bedroom at night; and of these, 62.5% had screens on when they fell asleep. Many children had at least one sibling that slept in the same room (60%) or the same bed (40%) with them at night. Most parents described their home as quiet (86.7%) and their neighborhood as safe from crime (93.3%). Sources and topics for child sleep guidance and areas parents struggle with most are listed in Table 2.

The CSWS indicated children resisted bedtime for an average of 48.7 ± 44.3 min/night and took 62.3 ± 52.7 min to fall asleep. Children woke an average of 2.0 ± 1.1 times/night and took 30.9 ± 62.8 min to fall back asleep after waking. In the morning, children took an average 29.4 ± 38.5 min to become alert. Values for CSWS subscales averaged 2.5 ± 0.9 (going to bed), 3.1 ± 1.1 (falling asleep), 3.9 ± 1.1 (maintaining sleep), 3.9 ± 1.1 (reinitiating sleep), and 3.4 ± 0.8 (returning to wakefulness). Total sleep quality scores averaged 3.4 ± 0.7. For context, another sample of good and average sleepers had total scores of 5.2 ± 0.4 and 4.3 ± 0.4, respectively [22], suggesting this study’s sample comprised lower-than-average sleepers. Median and IQR values for the CSWS variables in this sample are listed in Appendix A. 

### 3.2. Barriers to Optimal Child Sleep 

Six themes emerged from the qualitative data related to barriers to optimal child sleep. Each theme was mapped onto associated TDF domains, as described in Table 3, and indicated in parentheses next to the representative quotes below.

#### 3.2.1. Theme 1: Stimulating Bedtime Activities

When describing what activities were typically performed before bed, parents often mentioned stimulating activities. Most often, this included watching television as part of a bedtime routine or their child falling asleep when outside of the home while attending activities (e.g., a sibling’s baseball game). Some parents used television as a means of soothing their child or in response to their child’s behavior.


*“I’m terrible at routines. We could be out somewhere, and my son might fall asleep in the car on the way home and never wake up.”*
(Belief about capabilities)


*“Everybody’s TV is cut off, except for [my youngest child’s]. Because if her TV is not on, she will scream.”*
(Behavioral regulation)


*“He’s a little resistant to turning off the TV, so we’ll just turn it to a YouTube channel that has lullabies, and you know, the fish swimming in the water to try and calm him down.”*
(Behavioral regulation)

#### 3.2.2. Theme 2: Child Behavior Challenges

Behavioral challenges, such as child resistance or hyperactivity, were described as sources of variability in bedtime routines. During nighttime wakings, some parents also mentioned difficulty in getting their child back to sleep due to resistance in going back to bed. 


*“Sometimes [the siblings] feed off each other. Depending on how bad they are at it with each other, sometimes things can take longer… I can’t get them to focus on washing themselves.”*
(Belief about capabilities)


*“It’s like a negotiation to get him back to sleep.”*
(Behavioral regulation)

#### 3.2.3. Theme 3: Variability in Child’s Daily Structure

Parents described variability in daily structure as influencing child’s sleep schedules. Structure was considered pre-planned activities or established routines [36]. Sources of structure could be external influences (e.g., childcare) or the extent of routines within the home (e.g., varying rules when living between two households). At times, structure promoted optimal sleep (e.g., attending daycare); while at other times, planned activities hindered ideal sleep schedules (e.g., sports events late at night). Some parents seemed to have lower-priority goals and intentions around maintaining routines when structure varied (e.g., weekdays vs. weekends).


*“On the weekend, we sleep in. We lay around in our pajamas. On most days that we don’t have an activity planned, that’s what we do.”*
(Intentions)


*“If we had something else to do, like we went to the zoo or to my parents, and we’re coming home late in the afternoon, it’s a pretty much a done deal that he’s going to fall asleep.”*
(Goals)


*“[Naps are] definitely more consistent at daycare. At home, the range is a lot wider.”*
(Environmental context and resources)

#### 3.2.4. Theme 4: Parent Work Responsibilities

For some parents, work obligations contributed to varying sleep schedules, such as if a parent came home late and routines shifted. Parents’ work was also mentioned as a possible barrier to trying new sleep strategies given a lack of time. 


*“My husband and I are both entrepreneurs, so we work when we can. We still have work to do by the time he’s back from daycare, and sometimes that can change the routine.”*
(Professional/social role and identity)


*“For first time parents like us, who are also working, we don’t have the luxury to have new strategies…. If I was in a less demanding job, maybe I would have more time, and I would be more open to experimental strategies.”*
(Professional/social role and identity)

#### 3.2.5. Theme 5: Sleep Hindering Environment

Some parents described the bedroom and home environments as not promoting optimal sleep, such as high levels of noise or co-sleeping with their child. At times, this was environmentally driven with multiple children in the same bed or creating noise at night. Other times, parents reinforced co-sleeping, or felt they could not break a bedsharing habit.


*“It can sometimes get noisy having babies in the house… last night, they decided they were going to refuse to sleep at 1:00 am, and they were up screaming until 3:00 am.”*
(Environmental context and resources)


*“I can’t get [the siblings] to separate if I wanted to. The six-year-old goes to sleep fighting with the four-year-old…No matter what I do with separate beds…they always end up back together.”*
(Belief about capabilities)


*She’ll wake up and come sit on my lap for 10 to 20 min and fall back asleep. As long as she’s touching me, she’s fine.*
(Reinforcement)

#### 3.2.6. Theme 6: Parent’s Emotional Capacity

Parents expressed feelings of fatigue, stress, and frustration when managing their child’s sleep. These negative feelings often resulted in inconsistent routines or delayed bed timing.


*“If you’ve had a long day, and are out of gas, and don’t feel like you want to, you’re just not being consistent. That’s definitely a challenge.”*
(Emotion)


*“It depends on my day…I work in the heat. Sometimes I’m tired. Honestly, sometimes I come home and crash, and then I take care of him.”*
(Emotion)

### 3.3. Facilitators to Optimal Child Sleep 

When asked what strategies were useful and effective for managing child sleep challenges or what promoted greater sleep consistency on certain days, 4 themes emerged. These included: soothing activities, preparation and action planning, establishing routines and limits, and creating a soothing environment.

#### 3.3.1. Theme 1: Soothing Activities

Many parents incorporated some soothing strategies in bedtime routines and when their child experienced frequent night wakings, such as reading books, playing soft music, and rubbing their child’s back.


*“It’s probably around 9:30 or 10:00 by the time she feels sleepy, and we are done with our songs. After that, she asks for books. Then we read stories until she falls asleep.”*



*“I’ll put him back in bed, in the position that he usually falls asleep in, and I’ll maybe rub his back for a little bit.”*


#### 3.3.2. Theme 2: Preparation and Action Planning

Some parents used pre-planned strategies and tools, such as alarms, to try and promote routines for more consistent sleep and signal when certain activities should begin.


*“Pre-planning and having everything ready. Pajamas out and everything kind of lined up definitely helps move things along.”*



*“Setting an alarm on my phone or just being aware of the time and starting the routine at that time.”*


#### 3.3.3. Theme 3: Establishing Routines and Limits

Some parents created structured routines and established rules to minimize sleep challenges. Often this was motivated by children’s resistant behavior if routines were not in place. These strategies indicated parent’s knowledge of the benefit of routines and setting limits.


*“We’re trying to have a more consistent routine. Or the routine we have, just implementing it more consistently. I think that helps.”*



*“I developed a green light, yellow light, red light type thing. I put it close to where he was. He had consequences if he wouldn’t stay in his room or was being too loud. That seemed to help. If he stayed on green, the next day he got a little prize.”*


#### 3.3.4. Theme 4: Creating a Soothing Environment

While some parents mentioned natural sleep-promoting environments, such as quiet neighborhoods and minimal light, other parents used strategies to minimize environmental noise after their children went to bed, as to not disrupt their child’s sleep.


*“The neighborhood where we are located is really peaceful and quite.”*



*“[My husband] will watch something on his phone with his headphones in.”*


### 3.4. Motivations for Participating in a Child Sleep Intervention 

Parents reflected on factors that would encourage them to participate in a child’s sleep intervention. Three themes emerged, as described in Table 4. Parent’s current frustrations were expressed as motivation to try new intervention strategies. Parents felt a desire to quickly see tangible results when trying new approaches and felt this would provide strong reinforcement to continue with an intervention. Parents also indicated a strong preference to receive advice from, and participate alongside, other parents with similar struggles to where they could relate.

Parents reflected on the anticipated benefits of improving child sleep. Two themes emerged (Table 4) and included improved family-level sleep and emotional well-being. Parents felt their own sleep would improve, if they no longer experienced child sleep challenges, and this would result in greater alone time, marital dynamics, positive parent/child emotions, and family cohesion. 

### 3.5. Intervention Preferences

Survey data indicated most parents (80%) would be interested in participating in a child sleep intervention. When asked if other parents they know would be interested, 40% reported yes, while 46.7% were unsure, and 13.3% said no. Virtual delivery (rather than in-person) was preferred by 60% of parents, with mixed preferences for the group (47%) versus individual (53%) sessions. The most realistic amount of time most parents (80%) could devote to each intervention session was 30–45 min.

Qualitative themes were derived for parent preferences for a prospective intervention (Table 5). Parents preferred a virtual delivery (rather than in-person), as it was more convenient, and minimized barriers around transportation, childcare, work, and other time commitments. Those preferring group delivery cited the opportunity to converse with other parents in similar situations as the primary benefit, while a few shared that their discomfort in social settings would limit their interest in a group intervention. Parents who preferred individual delivery mentioned the opportunity for more customized content that could address their specific needs and concerns, as opposed to general content that may apply to a wider range of families.

## 4. Discussion

This study provided rich formative data on barriers and facilitators to optimal child sleep and parents’ preferences for behavioral sleep interventions. Prominent themes included sleep routines, environmental contexts, parents’ work and emotional capacity, and child-driven behaviors, which spanned 8 of the 14 TDF domains. Parents expressed a desire for future interventions to be relatable and produce tangible results, with motivation fueled by current frustrations and anticipated benefits in family-wide sleep and emotional well-being. Lastly, parents expressed an interest in virtual intervention delivery, given the ability to reduce barriers and provide flexibility. Collectively, this evidence can inform the development of prevention-focused sleep interventions among preschool-aged children for later-life disease prevention. 

A central aspect of promoting optimal sleep is establishing consistent bedtime routines, with soothing activities conducted in the same order each night [37]. Bedtime routines can result in improved sleep [38] and positive developmental outcomes, such as emotional and behavioral regulation and family functioning [37]. Many parents mentioned the inclusion of both stimulating and soothing activities as part of their child’s bedtime routine, such as television viewing and reading books, respectively. Television viewing before bed is common; yet, this has been shown to result in poor sleep, and subsequent child behavior problems [39]. Thus, evidence-based guidelines recommend setting positive routines, not allowing screens in the bedroom, and restricting screen viewing before bed [40,41]. Some parents indicated screens were used in response to behavioral resistance or hyperactivity, while other parents indicated this was part of their typical nightly routine. As such, behavioral interventions should consider promoting positive bedtime practices by empowering parents through the reinforcement of soothing activities before bed and strategies to address the intent behind family’s current use of stimulating activities, such as behavioral management.

Structured settings are thought to promote optimal sleep by regulating children’s bed and wake times [36]. Parents in this study felt that child sleep was more consistent when certain sources of structure were provided, such as days when children attended daycare or had planned activities. Wake times, nap schedules, and subsequent nighttime sleep seemed to improve when children had more regulated schedules. Contrary to this, weekends were often less regulated and included unscheduled activities that contributed to children being more likely to experience variable sleep patterns (e.g., irregular naps, sleeping in). At times, weekdays with planned extracurricular activities for older children (e.g., evening sport practices) resulted in less optimal sleep by delaying bedtime for their younger sibling. Shifts in sleep timing, such as large weekday-weekend differences, have been associated with less healthful dietary behaviors in older children and adolescents [42,43,44]. Thus, early childhood may be a critical time to establish positive, consistent patterns to prevent the development of associated negative health behaviors. Data from this study can inform interventions in supporting parents by setting intentions and implementing goals around maintaining consistent routines, to the extent possible, on days when structure naturally varies. For example, some parents mentioned time-driven tools, such as setting alarms, or prioritizing consistent routines to help with child sleep challenges. Tools such as these may be beneficial in maintaining consistent sleep routines (e.g., naps on weekends, bed and wake times on days when children do not attend childcare) to promote healthy sleep patterns and contribute to fewer sleep-related challenges. 

Parent’s feelings of stress and fatigue emerged as a prominent barrier influencing child sleep patterns. Parents’ mental health is bidirectionally linked to child sleep [45], such that worse psychosocial health influences child sleep [46], and child sleep difficulties, in turn, influence parents’ emotional well-being [47]. Parents mentioned stress and fatigue as rooted in external factors (e.g., work), as well as challenges around child sleep. They acknowledged that, if they were to no longer experience child sleep difficulties, they would anticipate beneficial improvements in their own sleep and psychosocial well-being. Given that low-resourced households are more likely to experience both higher levels of stress [48,49] and suboptimal child sleep [4], parents’ emotional health must be considered within family-based interventions to promote optimal child sleep. Such considerations of parental well-being could result in holistic family-wide benefits, such as more positive parenting [50], consistent bedtime routines [51], and improved child sleep. 

Many themes derived from this study align with previous work that has explored barriers and facilitators to optimal child sleep. For example, one study with focus groups among parents and their elementary-aged children also found that stimulating activities, such as technology, was a common barrier to later bedtimes [52]. For facilitators, consistent bedtime routines are beneficial for promoting sleep in all children, regardless of factors, such as family income or child age. In addition to findings that align with previous work, this study extends the current body of knowledge in numerous ways. First, barriers to child sleep were mapped onto TDF domains that directly inform subsequent intervention development. Interview data were used to derive situational context and parent intent to identify specific TDF domains that can inform evidence-based intervention strategies. Second, this formative work gathered intervention preferences and motivations from parents as key stakeholders. Third, younger children’s daytime schedules and napping needs are more variable than older children [7]. Thus, results from this study pertaining to daytime factors (e.g., daycare) and napping influences are novel. Lastly, future work in this study will investigate how themes in families with low income are similar and distinct from those in higher-income households.

Stakeholder perspectives and viewpoints from the target population have become increasingly prioritized in intervention design to foster relatable content and provide impactful outcomes [53]. This study sought to understand parents’ preferences for child sleep interventions, including the format, length, and mode of delivery. Results indicated variability in intervention preferences, though most parents felt virtual delivery offered greater benefits than in-person sessions. Digital interventions have become common in behavioral health to expand reach and reduce barriers to participation [54]. Parents in this study echoed this sentiment by mentioning virtual delivery would increase scheduling flexibility and reduce obstacles to their involvement (e.g., transportation). Preferences for individual versus group sessions were variable, indicating the need for interventions to offer multiple modalities that accommodate diverse circumstances and preferences. Benefits for group sessions included social support, while individual sessions were thought to provide tailored content and accommodate those with social anxiety in group settings. Collectively, parents seemed to prefer shorter intervention sessions (i.e., 30–45 min/session) which should be considered to enhance attendance and retention. Lastly, parents favored the idea of guidance coming from those who experience similar struggles, highlighting the importance of intervention delivery from relatable community members. 

This study involved parents who perceived current sleep challenges and desired guidance to improve their child’s sleep. As such, this sample likely has greater awareness of sleep-related issues and may be internally motivated for intervention participation. Results may not generalize to parents with less awareness of optimal child sleep patterns or to those who do not desire guidance. Conversely, families with known child sleep disorders or medical conditions that impacted their child’s sleep were excluded, which limits the exclusion criteria to those with prescribed medical conditions. Eligibility criteria did not specify how long families had to be experiencing child sleep challenges; thus, future research should examine if parents’ perceptions vary based on this factor. This study was strengthened by rigorous mixed methodology and the application of the TDF theoretical framework, designed to inform behavioral intervention development. The study population consisted of families with low income, which helps to address a knowledge gap in child sleep intervention research. 

## 5. Conclusions

Parents experienced multiple barriers to optimal child sleep in their preschool-aged children, which spanned behavioral, environmental, and emotional domains. Child sleep challenges impacted other family members’ sleep and emotional well-being; thus, parents were motivated to try new intervention strategies given current frustrations and the possibility for tangible results. Parents wanted future interventions that were relatable, with short sessions delivered virtually to minimize barriers and provide flexibility. This formative evidence will be used to support the development of future behavioral sleep interventions among families with low income to reduce sleep disparities and promote health equity for disease prevention. 

## Figures and Tables

**Table 1 ijerph-20-00862-t001:** Demographics of participating parents (*n* = 15) and their children.

	Parents	Children
Age, years (mean ± SD); median (IQR)	34.7 ± 8.2; 35 (8)	2.9 ± 0.8; 3 (2)
Female (%)	86.7	46.7
Race ^a^ (%)		
Black/African American	26.7	33.3
White/Caucasian	60.0	53.3
Other ^b^	13.3	26.7
Hispanic (%)	6.7	6.7
Marital status (%)		
Single	40.0	
Married	33.3	
Divorced	26.7	
Education (%)		
High school diploma/GED/or less	33.3	
Some college or vocational training	20.0	
Bachelors or Associates degree	33.3	
Graduate degree	13.3	
Employment ^a^ (%)		
Full time work	33.3	
Part time work	26.7	
In school	13.3	
Unemployed	40.0	
Annual household income, $ (mean ± SD); median(IQR)	30,000 ± 17,845; 29,000 (11,000)	
Insurance (%)		
Medicaid	80.0	
Private insurance	20.0	

GED = General Education Development Test; SD = standard deviation; IQR = interquartile range; ^a^ Response options were select all that apply. Values may sum to >100%; ^b^ American Indian/Alaskan Native, Asian, or reported “other” race.

**Table 2 ijerph-20-00862-t002:** Sources of guidance and areas parents struggle with the most related to child sleep. Data from *n* = 15 parents with a child 2–4 years of age.

	Parents (%)
Current sources of child sleep guidance ^a^	
Pediatrician or another doctor	60.0
Family member or friends	26.7
Websites (e.g., blogs, social media)	26.7
Child’s daycare	13.3
Topics parents received guidance about ^b^	
Bedtime routines	46.7
Naps	33.3
Bedroom environment	33.3
Night wakings	26.7
Nighttime sleep duration	20.0
Managing sleep challenges	20.0
Wakeup routines	13.3
Parents endorsing areas of struggle	
Time to fall asleep at night	46.7
Going to sleep at an appropriate time	46.7
Frequent waking in the night	46.7
Child sleeping in their own room	33.3
Trouble waking in the morning	33.3
Taking naps	26.7
Getting long enough sleep	20.0
Electronic use at bedtime	13.3

Response options were select all that apply. Values may sum to >100%; ^a^ One parent selected “other” and reported books; ^b^ One parent selected “other” and reported night weaning.

**Table 3 ijerph-20-00862-t003:** Qualitative themes on barriers to optimal child sleep and associated domains from the Theoretical Domains Framework (TDF). Descriptions of how TDF domains relate to themes are provided. Data were from *n* = 15 parents in low-resourced households, with a child 2–4 years of age, who struggle with their child’s sleep.

Themes	TDF Domain	Domain Description
**Stimulating bedtime activities**—Inclusion of stimulating activities within bedtime routines	Beliefs about capabilities	Parents felt they were not good at setting consistent bedtime routines
	Behavioral regulation	Stimulating activities (e.g., television) were used control child behavior before bed
**Child behavior challenges**—display of behavioral resistance or hyperactivity	Belief about capabilities	Parents feeling unable to control child’s behavior or get their energy out
	Behavioral regulation	Negotiating with children during night wakings to get them to go back to sleep
**Variability in child’s daily structure**—day-to-day changes in pre-planned activities or routines, such as part-time childcare, weekends, or extracurricular activities	Intentions	Intentional plans that result in child staying up late or sleeping in on less structured days (e.g., weekends)
	Goals	Goal priorities that do not promote consistent sleep when structure varied (e.g., doing activities, rather than keeping consistent mid-day naps on weekends)
	Environmental context and resources	Environmental changes (e.g., part-time childcare, living between two households) with varying structure, that resulted in inconsistent sleep
**Parent work responsibilities**—workload and work schedules	Professional/social role and identity	Parent’s workload and schedules impacted the timing and duration of child sleep
**Sleep hindering environment**—stimulating sleep environments, such as noise or bedsharing	Environmental context and resources	Stimulating environments contributed to insufficient sleep, such as neighborhood noise or bedsharing due to limited resources
	Beliefs about capabilities	Parents feeling unable to separate siblings from bedsharing, due to sibling’s desire to sleep together
	Reinforcement	Parents reinforcing less quality sleep by allowing children to bedshare with them when children want
**Parent’s emotional capacity**—feelings of stress and fatigue	Emotion	Parent’s stress, frustration, and exhaustion impacted child sleep

**Table 4 ijerph-20-00862-t004:** Qualitative themes on parent’s motivations for participating in a child sleep intervention. Data from *n* = 15 parents in low-resourced households, with a child 2–4 years of age, who struggle with their child’s sleep.

Themes	Description	Representative Quotes
**Motivations for participating in a child sleep intervention**		
Relatable situation	Parents desired receiving guidance from others in a relatable situation who understand their experiences	*“I find that it’s always harder to take someone’s advice when they do not know what you’re going through.”*
Seeing tangible results	Parents felt a strong motivator for trying new strategies would be to see tangible improvements	*“I think when you can see the benefit. If you can see where you’re getting more sleep as a parent, or you’re getting to relax at night… I think that would be a huge benefit to see.”*
Current frustrations	Parents current frustrations would encourage them to seek out an intervention to try and improve their child’s sleep	*“I have a 20-year-old and a 17-year-old, and along comes this 4-year-old. Nothing I did with [the older children] works. It’s very frustrating… It’s like, gosh, I just want to find something that works. That would motivate me to try some new stuff.”*
**Anticipated benefits of improving child sleep**		
Improved family member’s sleep	Parents felt their sleep, and sibling’s sleep, would improve if their child were to no longer experience sleep challenges	*“If I didn’t get woken up in the middle of the night, I would be able to sleep better. I think I would be more rested in the morning.”*
Improved family emotional well-being	Parents felt emotional well-being would improve, such as alone time, marital dynamics, and parent/child emotions	*“If he would sleep better, I would have a lot better start to my day. I would be able to have more alone time in the morning, which is obviously life giving to me when I’m giving my whole days to my kids.”*

**Table 5 ijerph-20-00862-t005:** Qualitative themes on parent’s preferred mode of delivery in a child sleep intervention. Data were from *n* = 15 parents in low-resourced households, with a child 2–4 years of age, who struggle with their child’s sleep.

Themes	Description	Representative Quotes
**Virtual delivery**		
Flexibility and convenience	Virtual delivery was described as more convenient and accommodating to schedules	*“I think the virtual environment is definitely easier this day and age for people to schedule around.”*
Minimizes barriers	Virtual delivery was thought to minimize barriers to participation	*“I would prefer it to be virtual, so I’m not having to get in my car and drive somewhere. You know, gas prices going up like crazy.”*
**Group sessions**		
Social support	Group sessions would provide peer support from other parents	*“A group setting would be better… hearing other parents’ struggles or other successful strategies they’ve implemented. That would be helpful to me along with, of course, the instructor.”*
**Individual sessions**		
Personalization	Individual sessions were described as more personalized to parent’s needs	*“I would want to work with someone individually… my kid has very specific sleep issues for our family… I don’t want to be in a class where I have to hear all those basics… I really just want somebody to work with his specific problems if I’m going to devote time to that.”*
Social anxiety	Some preferred one-on-one settings, due to anxiety being in groups	*“I don’t like doing groups because I don’t like talking in front of people.”*

## Data Availability

The data presented in this study are available on request from the corresponding author.

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
