# Peer review of "Barriers to Optimal Child Sleep among Families with Low Income: A Mixed-Methods Study to Inform Intervention Development"

_ijerph, 2023, doi:10.3390/ijerph20010862_

Round 1

Reviewer 1 Report

The manuscript investigated perceived barriers to child sleep among parents with low income using a mixed-method approach. The study is interesting and promising to serve as a fundamental study to develop future interventions for improving child sleep. At this moment, I have several suggestions and/or concerns about the manuscript.

1.      In line 88, the authors indicated that they recruited 15 parents between March and June 2022. I wondered if there was any sample size calculation prior to recruitment or if it depended on how many parents the research team could recruit. Any influence of the COVID pandemic on parents’ behaviors or child sleep? Would it be possible to identify how long a child had trouble sleeping while recruiting?

2.      Section 3.1: I could find those numbers in neither Table 1 nor Table 2. Would those numbers come from any supplementary Tables? In addition, in lines 191-192, would that mean another sample and the total scores from another sample were compatible with the sample in this study? That is, the sample in this study had poorer sleep quality?

3.      In line 143, the authors reported means for continuous variables. Would it be more appropriate to report medians due to the sample size?

4.      Table 1: for private insurance, would this be those parents with any private insurance and with Medicaid; or with only private insurance?

5.      Section 3.2: I would suggest using the same order of TDF domains in parentheses within each theme as in Table 3. For example, (belief about capabilities) in line 211 was first mentioned in Table 3 so it would be easier to read when it is moved before (behavioral regulation) in line 206.

6.      Table 3: I am not sure if this suggestion should direct to the authors or editors, but I would suggest moving Variability in child’s daily structure in Themes up for one cell to align with Intentions in the TDF domain in order to make it clearer that the Intentions belong to the variability theme. Similarly, Sleep hindering environment could move up next to Environmental context and resources.

7.      Section 3.5: I could not find the numbers mentioned in lines 325-330. Were they in any supplementary files?

8.      In line 392: it seems that there is a typo after “acknowledged that, f they”.

9.      I am curious if the parents perceived sleep challenges and child sleep interventions differ between low-income parents and other samples in the existing literature (e.g., parents with higher income, children aged 3-5 (preschool) or older, etc.)

10.   In lines 416-418, I guess the authors wanted to talk about a self-selection bias. However, these sentences sound more like strengths to me. It would be great to make it clearer in my opinion.  

Reviewer 2 Report

This study conducted a mixed-methods formative evaluation among 15 parents with low-income communities to identify perceived barriers and facilitators to optimal child sleep in preschool aged children.

The survey measures included items on demographics, the home environment, child sleep quality, current sleep guidance, and intervention preferences.

The qualitative study included semi-structured interview to understand parents’ perceived barriers/facilitators to child sleep, and motivations/preferences for future intervention delivery.

The authors indicate the importance of tailored approaches are needed that can identify the unique challenges and needs of families from low income.

The results from the study indicates various themes: stimulating bedtime activities, child behavior challenges, variability in children’s structure, parent work responsibilities, sleep hindering environment, and parent’s emotional capacity.

The study will benefit by identifying what are some of the unique barriers that are specific to these low income populations that have not been studied previously. The themes or the barriers identified seems applicable to all parents. The study has further limitations with the number of participants (15) included to draw any conclusions.

The results from the study has the potential to inform future work to develop a tailored parenting sleep intervention, based on the format and length and mode of delivery of the intervention. However variability among parent preferences would be good to explore if these preferences varied due to where the parents live or based on the education level, etc., however because of the small sample size, it may not be feasible to tease apart these differences.

Minor typographical errors lowered the enthusiasm for this study. In addition, the study does not add anything new to the literature of sleep disparities. The results indicate that for promoting optimal sleep, establishing consistent bedtime routines, promoting positive bedtime practices through soothing activities, parent’s emotional health environmental contexts, parents’ work, are important factors to consider for future interventions.  However, these findings are not novel for this population and have been reported by researchers previously. The authors need to highlight what is novel in the findings of their research. What are the factors at various levels that might be playing a role in sleep health disparities in this population?

This article may not be a good fit for the special issue that features research on the associations between sleep, diet and cardiometabolic risk, as the study has a very limited focus on exploratory formative research on barriers/facilitators to optimal sleep among children from low income families. There is no discussion of sleep, diet and its relevance to cardiometabolic risks.
